# Effects of Reducing Agent on the Activity of PtRu/Carbon Black Anode Catalyst of Direct Methanol Fuel Cell

**Yu-Wen Chen *** and **Han-Gen Chen**

Department of Chemical Engineering, National Central University, Jhongli 32001, Taiwan; tsubomi0915@gmail.com
* Correspondence: ywchen@cc.ncu.edu.tw

**Abstract:** A series of PtRu/carbon black catalysts were prepared by means of deposition-precipitation and reduced by various reducing agents. $NaBH_4$, HCHO and $NaH_2PO_2$, respectively, were used as the reduction agents. Some of the samples were reduced by various amounts of $NaH_2PO_2$ to investigate the effects of P/Pt ratios on the characteristics and activity of the catalyst. These catalysts were characterized by X-ray diffraction and transmission electron microscopy. The components of these catalysts were detected by X-ray fluorescence, X-ray photoelectron microscopy, and extended X-ray absorption of fine structures (EXAFS). The methanol oxidation ability of the catalysts was tested by cyclic voltammetry measurement. The results show that $NaH_2PO_2$ could effectively reduce the particle size of PtRu metal. It can suppress the growth of metal particles. In addition, the P/Pt ratio is crucial. The catalyst reduced by $NaH_2PO_2$ with a P/Pt ratio of 1.2 had the highest activity among all catalysts. It had the higher Pt and Ru metal contents and smaller metal particle size than the other catalysts. Its activity was 253.12 A/g, which is higher that than the commercial catalyst (Johnson Matthey H10100, 251.32 A/g).

**Keywords:** carbon black; fuel cell; PtRu alloy; anode catalyst; direct methanol fuel cell



## 1. Introduction

Direct methanol fuel cells (DMFC) have received increasing attention in the last few decades [1–4]. PtRu is used extensively as an anode catalyst. Since very high metal loading is used, the particle size of PtRu is large. Obtaining a small particle size of PtRu is a critical issue in applications [4–12]. Yao et al. [13] used hollow porous platinum-silver double-shelled nanocages for the electro-oxidation of methanol. The material possesses numerous surface defects and exposed catalytically active sites, resulting in high activity. Li et al. [14] synthesized concave PtCo nanocrosses for methanol oxidation reaction. The concave cross-like structure can provide more under-coordinated atoms as active sites and shows great enhanced MOR performance than commercial Pt/C. Li et al. [15] recently reported a review on the roles and mechanism of various types of amino-based molecules which can be used to control the morphology of the noble metal crystallites and results in high electrocatalytic activity. Carbon black has been used as the anode of fuel cells. Vulcan XC-72 and black Pearl 2000 (from Cabot), acetylene black (from Denka and Shawinigan Black), and Ketjen Black (from Ketjen) have been used in practice. After loading PtRu on carbon black, the reducing agent is also crucial. Various reducing methods have been reported in the literature [13–19]. Xue et al. [19] reported that the PtRu/carbon black reduced by $NaH_2PO_2$ resulted in the deposition on the surface of catalyst. It decreased PtRu metal particle size, increased metal surface area, and resulted in high activity for the methanol oxidation activity and enhanced the oxidation activity of carbon monoxide.

In the present work, platinum-ruthenium alloy supported on carbon black by various reducing agents were studied. In this study, carbon black with a large surface area (855 m$^2$/g) was used as the substrate. PtRu was loaded by means of the deposition-precipitation method and reduced by various agents. The sample was characterized in

detail. Single cell measurement was used to determine the oxygen reduction reaction (ORR) activity.

## 2. Experimental

### 2.1. Chemicals

Chloroplatinic acid hydrate was obtained from Aldrich (Saint Louis, MO, USA). Ruthenium trichloride hydrate was purchased from Sigma-Aldrich (Darmstadt, Germany), ethanol was purchased from Sigma-Aldrich (Darmstadt, Germany), hydrochloric acid and sulfuric acid were purchased from Fisher (Hampton, NH, USA), and Nafion Solution was obtained from Dupont (Wilmington, DE, USA). Carbon black was obtained from Ketjen (ECP300JD, Tokyo, Japan)). The commercial catalyst was obtained from Johnson Matthey (H10100, London, UK) for comparison (60 wt.% PtRu/C). Hydrogen and nitrogen gases were purchased from Air Products (Allentown, PA, USA).

### 2.2. Preparation of Anode Catalyst

In this study, carbon black from Ketjen (ECP300) was chosen as the support for PtRu metal. Pt-Ru/carbon black catalyst was prepared by the following method. $RuCl_3$ and $H_2PtCl_6$ metal salts were dissolved in distilled water simultaneously. Carbon black support was then added in the above solution. The solution was agitated for 1 h. The reducing agents, such as $NaBH_4$, HCHO and $NaH_2PO_2$, respectively, were added into the above solution. It was washed by distilled water and dried in a vacuum oven. These samples were denoted as PtRu/carbon-m, where m is the reducing agent. Some of the samples were reduced by various amounts of $NaH_2PO_2$ to investigate the effects of P/Pt ratios on the characteristics and activity of the catalyst. The catalyst reduced by $NaH_2PO_2$ was denoted as PtRuP/carbon (x), where x is the atomic ratio of P/Pt in the starting materials.

### 2.3. Characterization of Catalysts

The XRD pattern of each sample was recorded on a diffractometer (PANALY X'PERT, Philips, Amsterdam, Holland) with Cu-$K_\alpha$ radiation. The sample was scanned from $10°$ to $85°$ at a scanning rate of $0.025°\ s^{-1}$.

The surface area (Sg), pore volume (Vp), and pore size distribution of catalysts were measured by the ASAP2400 (Micromeritics Instrument Corporation, Norcross, GA, USA) surface area and pore size analyzer. $N_2$ physisorption-desorption isotherms were obtained at $-196\ °C$ by the BET equation for surface area and the BJH method for pore size distribution calculations.

TPD was used to monitor the oxygen-containing functional groups on the surface of carbon black. At high temperatures, the functional groups decomposed to release CO and $CO_2$. Based on the TPD peaks, one can determine the acidity of functional group on the surface.

A 0.5 g sample was loaded in a quartz U tube reactor. Helium was used as the carrier gas. The temperature of reactor was raised to $1000\ °C$ with a heating rate of $5\ °C/min$. The outlet gas was sampled every 1 min and analyzed by a gas chromatograph equipped with a thermal conductivity detector (TCD).

Transmission electron micrographs (TEM), energy dispersion scanning (EDS) and selective area electronic diffraction (SAED) analyses were performed on a JEM-2000FX (Tokyo, Japan) instrument using an accelerating voltage of 200 kV.

Thermogravimetric analysis (TGA) was carried out (Mettler Toledo, Columbus, OH, USA) under dry air with flow rate of 40 mL/min. Temperature was raised from 25 to $950\ °C$ with a heating ramp of $10\ °C/min$.

X-ray photoelectron spectroscopy (XPS) was carried out with a ESCALAB 250 XPS (VG Scientific, Waltham, MA, USA). The XPS spectra were collected using Al K$\alpha$ radiation at a voltage and current of 20 kV and 30 mA, respectively. The base pressure in the analyzing chamber was maintained in the order of $10^{-9}$ Pa. The pass energy was 23.5 eV and the binding energy was calibrated by contaminant carbon ($C_{1S}$ = 284.5 eV). The peaks of each

spectrum were organized using XPSPEAK software; Shirley type background and 30:70 Lorentzian/Gaussian peak shape were adopted during the deconvolution.

X-ray fluorescence (XRF, Philips PW2400 Spectrometer, Amsterdam, Holland) was used to analyze the compositions of the samples.

The conductivity of carbon black was measured with a Four Point Probe Tester and Potentiostat (Autolab PGSTAT30).

X-ray absorption spectroscopy (XAS) of palladium was analyzed by using synchrotron radiation in fluorescence mode in the National Synchrotron Radiation Research Center, Hsinchu, Taiwan. The light source (TPS BL44A) provided photon energy ranging from 4.5 to 34 keV, which can be adjusted by a Si(111) monochromator. The oxidation state of palladium was also acquired using a linear combination referring to the standards of Pd foil, $PdCl_2$, and $PdO_2$, and the fit of the XANES spectra using Athena software. In addition, the coordination of the first core shell was analyzed using FEFF calculation by the Artemis interface depending on a secondary electron backscattering mode based on the EXAFS data. Pd K-edge steps were normalized to unity, and the data above Pd K-edge was background subtracted to produce the $\chi(k)$ functions, then, converting from the energy (eV) to k space over the region from 2 to 15 $\text{Å}^{-1}$. $k^3$-weighted $\chi(k)$ data, we proceeded with Fourier transformation to form a partial radial distribution function (RDF) around Pd and used a Hanning termination window of 2.0 $\text{Å}^{-1}$ at both ends of transformation range.

### 2.4. Methanol Oxidation Activity

The methanol oxidation activity of PtRu/carbon black was measured with a half cell. The detailed measurement procedure was reported in a previous paper [4]. Briefly, PtRu/carbon black catalyst was added in a suitable amount of distilled water to form slurry. The catalyst slurry was dropped on a rotating electrode disk. The sample was dried at 50 °C for 50 min. Nafion in isopropanol solution was then added in the catalyst and dried in nitrogen atmosphere.

Catalyst ink was prepared by dispersing the electrocatalysts in methanol with agitation. It was then dropped onto a glassy carbon surface. The Nafion alcoholic solution was dropped on the electrode surface and heated at 60 °C. The electrode was pretreated to remove surface contamination by cycling the electrode potential between 0 and 1.0 V vs. RHE at 50 mV s$^{-1}$ for 100 cycles in 0.5 M $H_2SO_4$. Electrocatalytic methanol oxidation was measured by chronoamperometry in 1 M $CH_3OH$ mixed with 0.5 M $H_2SO_4$. Catalytic activity was calculated by the current density at 0.5 V obtained by the CV method divided by PtRu catalyst loading.

## 3. Results and Discussion

### 3.1. Characteristics of Carbon Black

3.1.1. TPD of Functional Groups on the Surface of Carbon Black

Figure 1 shows the presence of oxygen-containing functional groups on the carbon black. The TPD peak at low temperatures was mainly from the strong acidic functional groups, and the desorption at high temperatures was from weak acidic functional groups [20,21]. This shows that the carbon black has both strong and weak functional groups on the surface. The amount of weak acidic functional groups was higher than that of the strong acidic function groups. The amounts of functional groups on carbon black are listed in Table 1.

**Table 1.** Conductivity and the amounts of functional groups on carbon black.

| Sample | [O] (mmol/g) | | | Conductivity (S/cm) |
|---|---|---|---|---|
| | CO | CO$_2$ | Total | |
| Carbon black | 0.4915 | 0.0905 | 0.5820 | 12.11 |

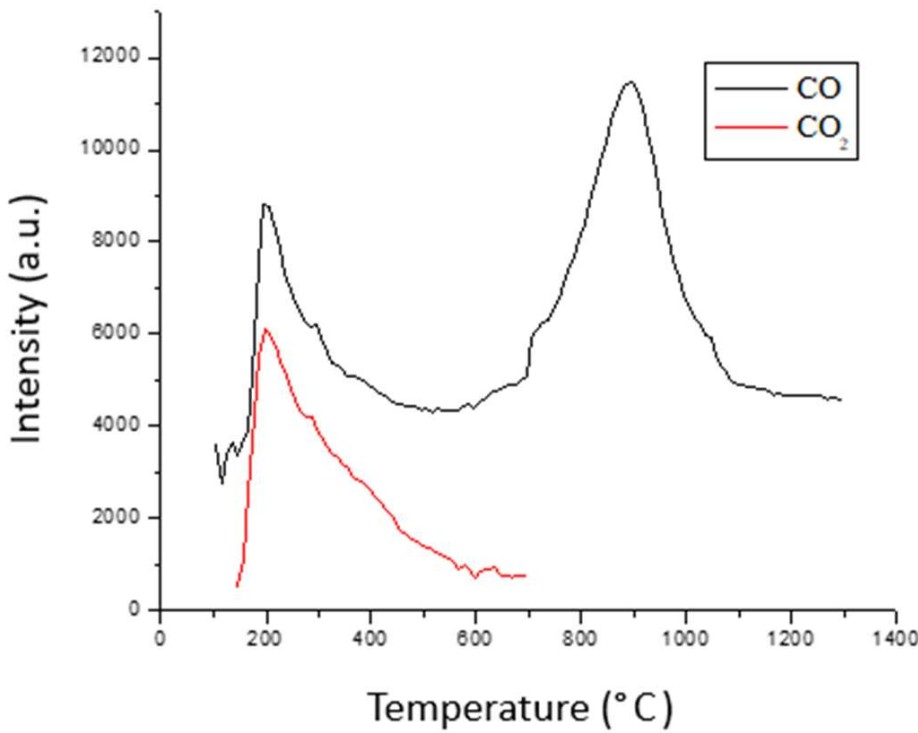

**Figure 1.** TPD profile of CO and $CO_2$ on the carbon black.

### 3.1.2. BET Surface Area and Pore Size Distribution of Carbon Black

BET surface area was calculated using the $N_2$ sorption isotherm, and pore size distribution was calculated using the BJH method. It shows that the surface area of carbon black was 855 $m^2$/g. The average pore diameter was 59 Å, and the total pore volume was 1.25 $cm^3$/g. The conductivity of carbon black was 12.11 S/cm, as shown in Table 1.

### *3.2. Characterization of PtRu/Carbon Black Anode Catalyst*
### 3.2.1. XRD

Pt is a face-centered cubic crystal. It has XRD peaks at 39.9°, 46.3° and 67.45°, corresponding to face (111), (200), and (220), respectively, in Pt/carbon black. In PtRu/C catalyst [22–24], since the atomic radius of Ru is smaller than that of Pt, the spacing of PtRu alloy crystal is smaller than that of Pt, resulting in the shift of XRD peaks to a higher degree.

Figure 2 shows the XRD patterns of the PtRu catalysts supported on the carbon black and reduced by various reducing agents; the Pt-Ru/carbon-$NaBH_4$ sample was reduced by $NaBH_4$, the Pt-Ru/carbon-HCHO sample was reduced by HCHO, and the Pt-Ru/carbon-$NaH_2PO_2$ sample was reduced by $NaH_2PO_2$. Scherrer's equation was used to calculate the crystallite size of the sample based on the peak intensity of phase (220). The crystallite of the Pt-Ru/carbon-$NaBH_4$ sample was 2.57 nm long. We were not able to calculate the crystallite sizes of the Pt-Ru/carbon-HCHO and Pt-Ru/carbon-$NaH_2PO_2$ samples, since the noise of the signal was too large and the peak intensity of phase (220) was too small. The results show that the sample reduced by $NaBH_4$ had the large metal particle size. The XRD peaks for Pt slightly shifted to a high angle, but this was not significant. There were no XRD peaks for Ru and $RuO_2$, inferring that Ru particles were too small to be detected.

Figure 3 shows the effect of the amount of $NaH_2PO_2$ on the properties of the catalysts. The catalyst is denoted as PtRuP/carbon (x), where x is the atomic ratio of P/Pt. As the amount of $NaH_2PO_2$ increased, the (200) peak and (220) peak could not be differentiated. The XRD peak of (220) became broader as the amount of $NaH_2PO_2$ increased, indicating that the particle size of the catalyst decreased. The particle size of PtRu/carbon (0.8) increased, possibly because the amount of $NaH_2PO_2$ was not enough to suppress the

growth of metal particles. When the amount of phosphorus increased to a P/Pt atomic ratio of 2, the XRD peak Pt (220) became narrow, indicating that the particle size of Pt was greater than the other samples. This is due to the large amount of P on the surface of carbon black, resulting in less space to adjust the metal particle size.

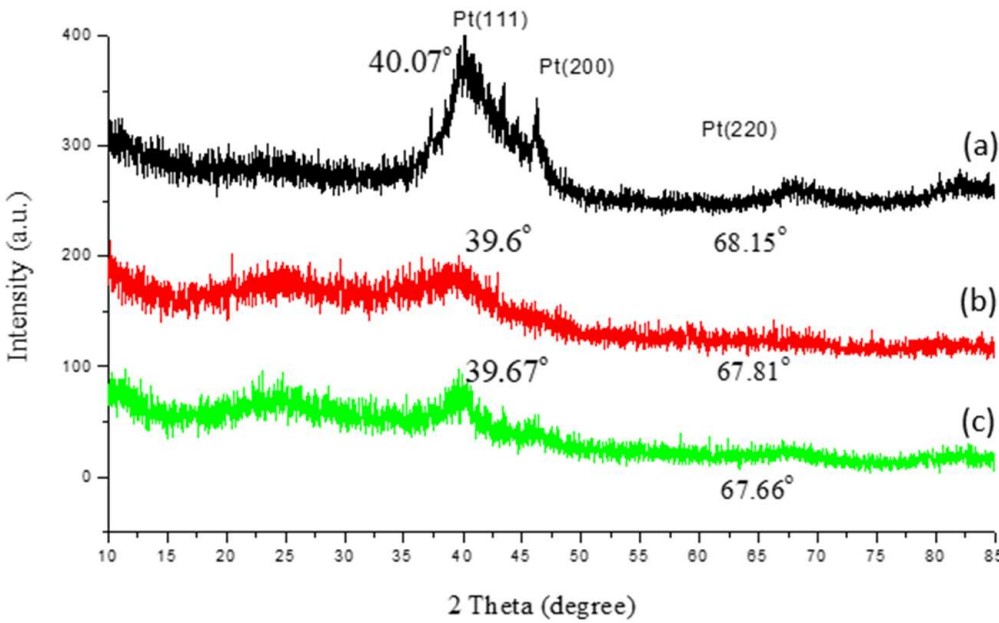

**Figure 2.** The XRD patterns of 60 wt.% PtRu supported on carbon black and reduced by various reductants. (**a**) PtRu/carbon-NaBH$_4$, (**b**) PtRu/carbon-HCHO, and (**c**) PtRu/carbon-NaH$_2$PO$_2$.

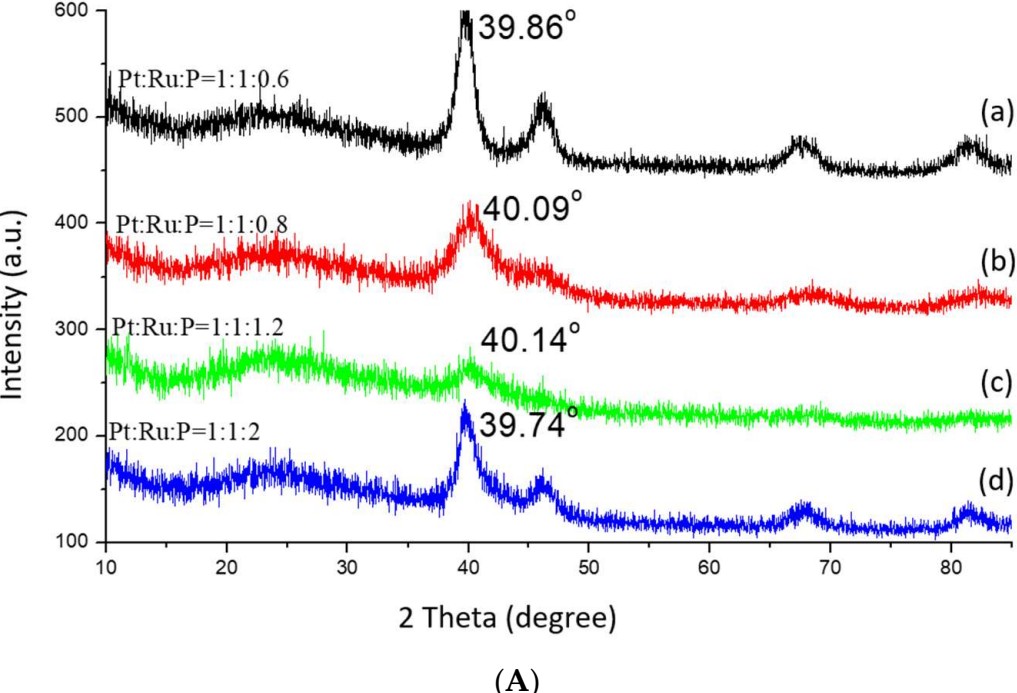

(**A**)

**Figure 3.** *Cont.*

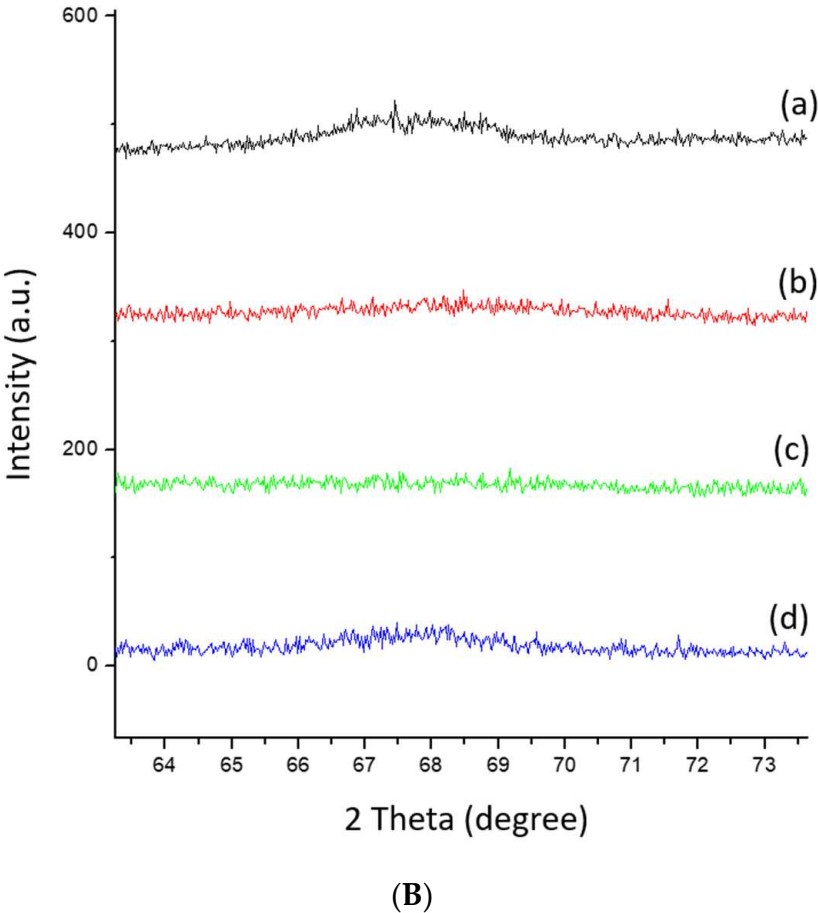

**(B)**

**Figure 3.** Catalysts containing 60 wt.% PtRuP/carbon black reduced with various amounts of NaH$_2$PO$_2$, (**A**) XRD patterns, (**B**) the comparison of the intensities of (220) XRD peaks. (a) PtRuP/carbon (0.6), (b) PtRuP/carbon (0.8), (c) PtRuP/carbon (1.2), (d) PtRuP/carbon (2).

### 3.2.2. TEM

The metal particle size of each sample is listed in Table 2. Figure 4 shows the TEM images of the samples reduced by various agents. The sample reduced by NaBH$_4$ had large particles and low metal dispersion. The sample reduced by HCHO had particles smaller than 3 nm, even though it has low amount of function groups. However, its PtRu metal dispersion is not high. The TEM image shows that it had some big particles. The sample reduced by NaH$_2$PO$_2$ had particles smaller than 3 nm. It has high metal dispersion. The results show that the sample reduced by NaH$_2$PO$_2$ can yield small particles and high metal dispersion.

**Table 2.** Effect of reducing agents on the particle size of the metal.

| Sample | Reduction | | | Particle Size (nm) | | Metal Dispersion |
|---|---|---|---|---|---|---|
| | Agent | T (°C) | Time (h) | TEM | XRD | |
| PtRu/carbon-NaBH$_4$ | NaBH$_4$ | RT | 2 | 3.5 | 2.57 | low |
| PtRu/carbon-HCHO | HCHO | 85 °C | 3 | 2.7 | N.D. | not good |
| PtRu/carbon-NaH$_2$PO$_2$ | NaH$_2$PO$_2$ | 90 °C | 10 | 2.59 | 2.03 | good |

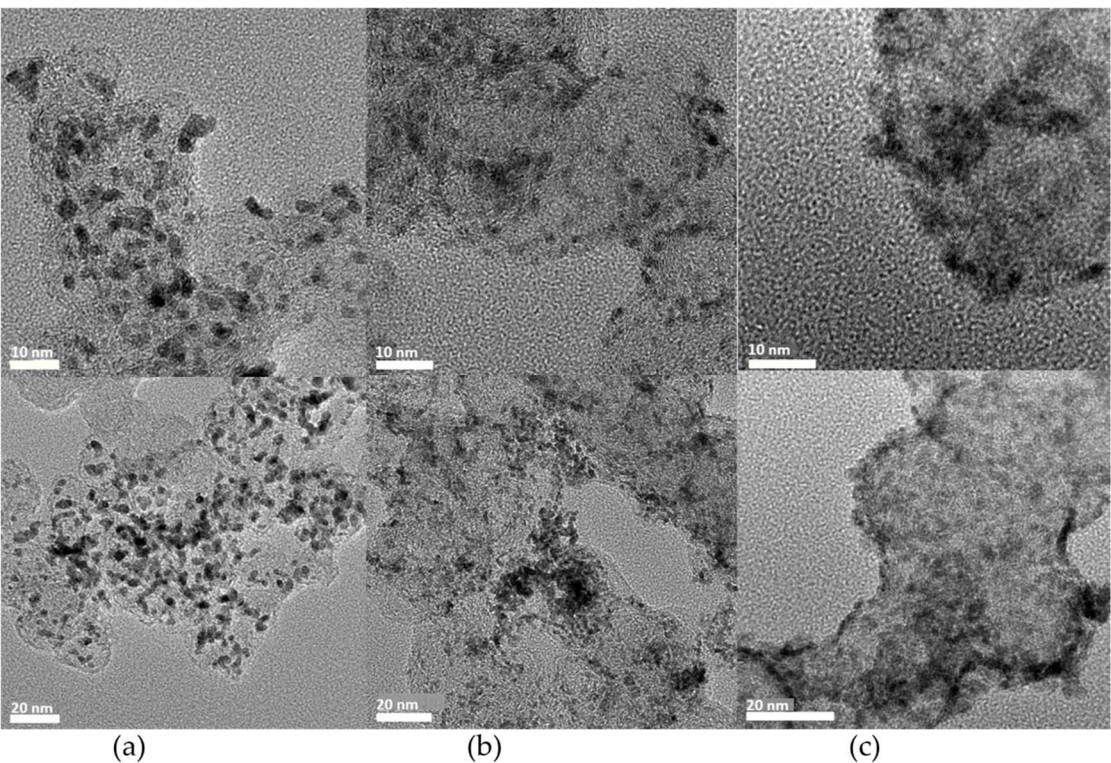

**Figure 4.** TEM images of the samples reduced by various reducing agents, (**a**) sample reduced by NaBH$_4$, (**b**) sample reduced by HCHO, and (**c**) sample reduced by NaH$_2$PO$_2$.

The difference in metal particle sizes is caused by the different reduction rates. The reduction rate by NaBH$_4$ is very fast. Metal cations were reduced before distribution homogeneously, and resulted in the non-homogeneous distribution of metal particles. The reduction rate by HCHO was slow, and it resulted in small metal particles. The reduction rate by NaH$_2$PO$_2$ was fast, but phosphorus could cover the surface of the carbon black [16]. It could thus regulate the metal particle size, and suppressed the growth of metal particles. Therefore, the metal particle was small and homogeneously distributed on the surface of the carbon black. In conclusion, NaH$_2$PO$_2$ is the best reducing agent among all samples in this study.

### 3.2.3. Effect of Phosphorus Content in Reduction on the Activity of the Catalyst

Some samples were reduced by various amounts of NaH$_2$PO$_2$ and resulted in various amounts of P in the sample [19]. Table 3 and Figure 5 show that the metal particle size decreased with increasing phosphorus content in the reducing agent. However, when the P/Pt atomic ratio was 2, the metal particle size increased. When the P/Pt atomic ratio was less than 1, phosphorus content was not high enough to suppress metal growth. Figure 6a shows that as the phosphorus content increased, the number of big particles decreased, and the number of small particles increased. When the P/Pt ratio was 1.2, it had the smallest metal particles among all samples. Figure 6d shows that the sample with a P/Pt ratio of 1.2 had a very uniform particle size distribution.

**Table 3.** Effect of phosphorus content in reducing agent on the metal particle size.

| Sample | P/Pt (Atomic Ratio) | Metal Particle Size (nm) |
| --- | --- | --- |
| PtRu/carbon (0.6) | 0.6 | 3.37 |
| PtRu/carbon (0.8) | 0.8 | 2.26 |
| PtRu/carbon (1.2) | 1.2 | 2.78 |
| PtRu/carbon (2) | 2 | 3.06 |

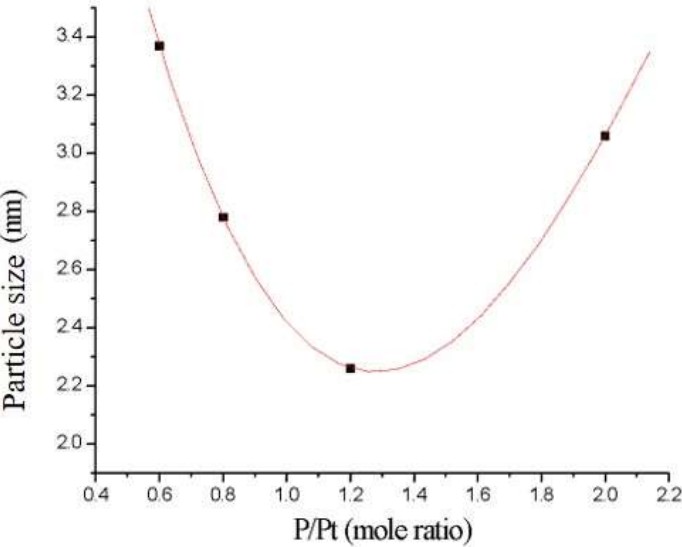

**Figure 5.** Metal particle size as a function of the amount of phosphorus used in the reduction process.

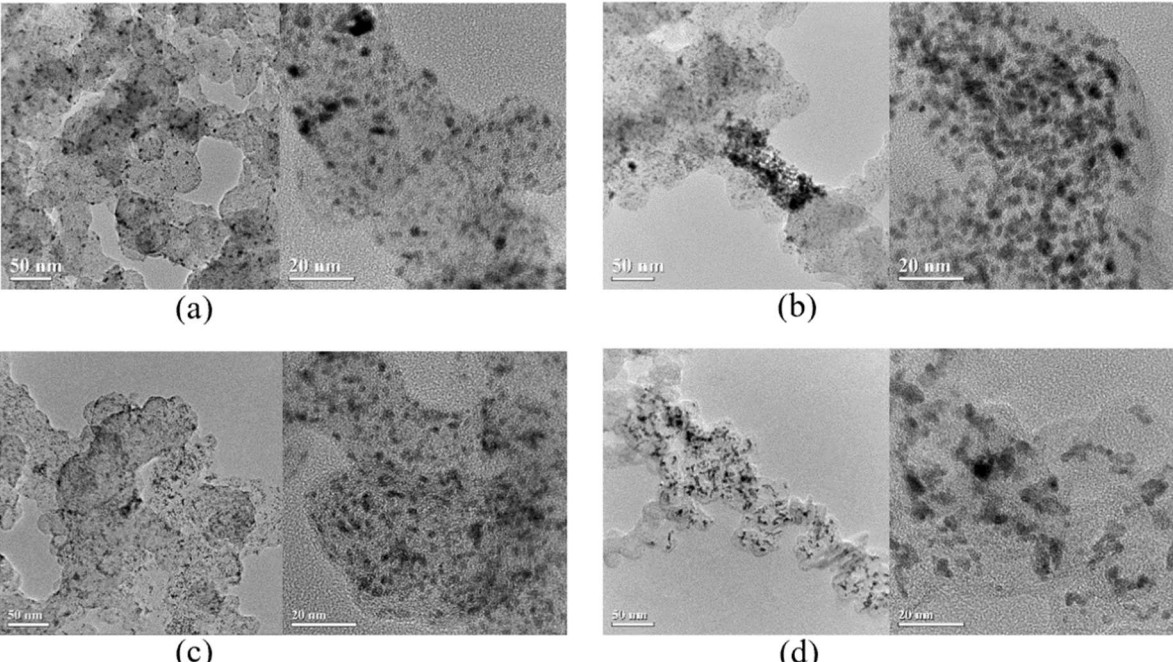

**Figure 6.** TEM images of the samples reduced by $NaH_2PO_2$ with various amounts of phosphorus in the reducing agent. (**a**) PtRu/carbon (0.6) (**b**) PtRu/carbon (0.8), (**c**) PtRu/carbon (1.2), and (**d**) PtRu/carbon (2).

### 3.2.4. TGA and XRF

One can obtain the metal loadings of PtRu on the carbon black from TGA. The weight loss between 0 and 200 °C was due to the adsorbed moisture on the sample. The weight loss between 200 and 500 °C was due to the oxidation of metal salts. The final product of TGA was analyzed by XRD. The XRD patterns show that the peaks did not shift compared to those of Pt/carbon black. This confirms that Pt was in a metallic state on the surface of carbon black, and Ru particles were small.

The metal Pt and Ru metal loadings from TGA and XRF analysis are listed in Table 4. The overall real metal loadings are close to the nominal loadings.

**Table 4.** TGA results of PtRu samples supported on carbon black and reduced by various amounts of NaH$_2$PO$_2$.

| Sample | P/Pt (Atomic Ratio) | Metal Oxide Content (wt.%) | XRF (mol. %) | | Loading (wt.%) | |
|---|---|---|---|---|---|---|
| | | | Pt | Ru | Nominal | Real |
| PtRuP/carbon (0.6) | 0.6 | 64.01 | 54 | 46 | 60.08 | 57.92 |
| PtRuP/carbon (0.8) | 0.8 | 67.75 | 58 | 42 | 60.01 | 59.08 |
| PtRuP/carbon (1) | 1 | 62.05 | 47 | 53 | 60.01 | 55.91 |
| PtRuP/carbon (1.2) | 1.2 | 62.15 | 52 | 48 | 60.03 | 56.15 |
| PtRuP/carbon (2) | 2 | 65.9 | 52 | 48 | 59.59 | 59.55 |

### 3.2.5. XPS

XPS was used to analyze the surface compositions and electronic states of the samples [19–21,23]. The XPS spectra of the samples are shown in Figure 7.

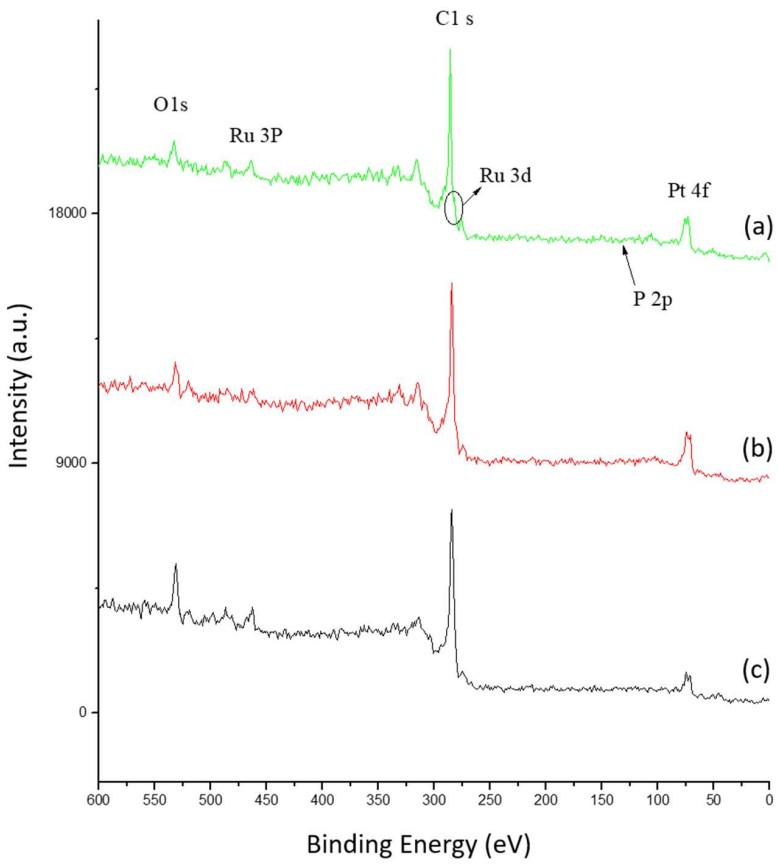

**Figure 7.** XPS spectra of the PtRuP/carbon catalysts reduced by NaH$_2$PO$_2$ with various phosphorus contents. (**a**) PtRuP/carbon (2), (**b**) PtRuP/carbon (1.2), and (**c**) PtRuP/carbon (0.6).

Table 5 lists the compositions of three samples with various phosphorus contents. PtRuP/carbon (0.6) had a Pt:Ru:P ratio of 1:1:0.6; PtRuP/carbon (1.2) had a ratio of 1:1:1.2; and PtRuP/carbon (2) had a ratio of 1:1:2. Figure 8 shows that the Pt 4f peak of the PtRuP/carbon (1.2) sample was higher than those of PtRuP/carbon (0.6) and PtRuP/carbon (2), indicating that this PtRuP/carbon (1.2) sample had a high content of Pt. The PtRuP/carbon (0.6) sample had the lowest peak area of Pt 4f among all the samples, indicating that it had the lowest Pt content. The peak area of O 1s of the PtRuP/carbon (0.6) sample was the highest among the three samples, and it had the highest oxygen content. All three samples did not have the peak of P 2p, inferring that the P content was very small.

**Table 5.** Effect of phosphorus content on the surface compositions (%) of the catalysts.

| Sample | Pt 4f$_{7/2}$ | | | Ru 3d$_{5/2}$ | |
|---|---|---|---|---|---|
| | 71.2 eV | 72.5 eV | 73.8 eV | 280 eV | 281 eV |
| PtRuP/carbon (0.6) | 54.79% | 45.21 | 0 | 0 | 100% |
| PtRuP/carbon (1.2) | 60.04 | 22 | 0 | 100 | |
| PtRuP/carbon (2) | 53.50 | 26.66 | 19.85 | 24.55 7 | 75.45 |

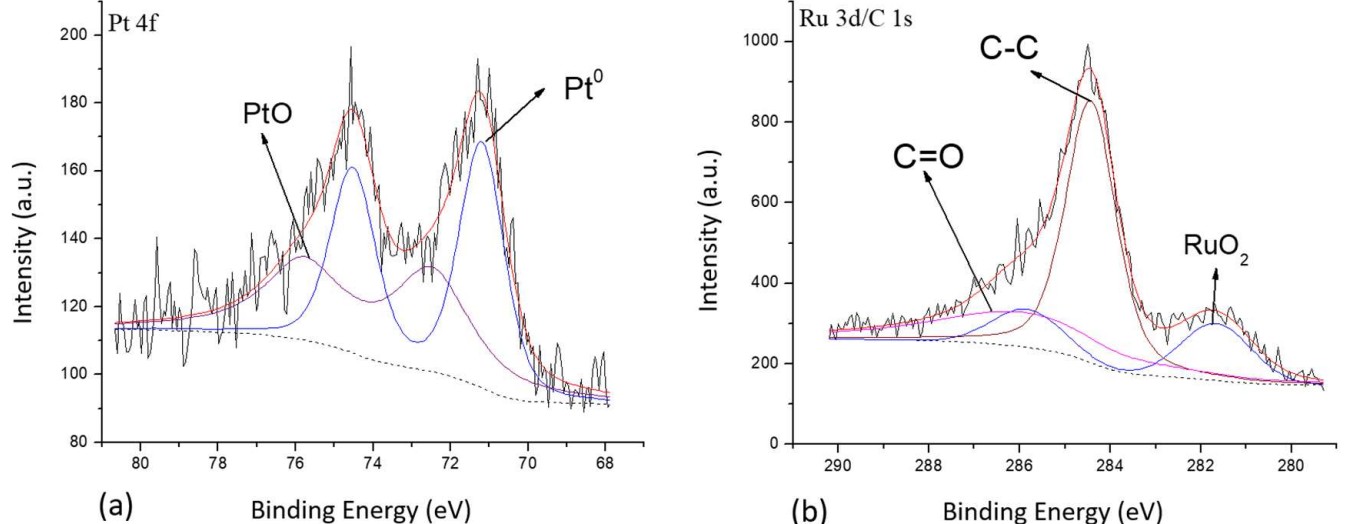

**Figure 8.** XPS spectra of the PtRuP/carbon (0.6) sample; (**a**) Pt 4f spectra; (**b**) Ru 3d/ C 1s spectra.

In the literature [19], it shows that Pt-Ru-P had interactions among the three elements, and the binding energy of Pt 4f became higher. The theoretical value of binding energy of Pt 4f is 71.1 [20–24]. The binding energy of Pt 4f$_{7/2}$ at 71.2 eV was assigned to Pt$^0$, that at ~72.5 eV was assigned to PtO, and the peak at ~73.8 was attributed to PtO$_2$. The binding energy of Pt 4f$_{7/2}$ in the PtRuP/carbon (0.6) sample was 71.2 eV (Figure 8), PtRuP/carbon (1.2) was 71.3 eV (Figure 9), and PtRuP/carbon (2) was 71.2 eV (Figure 10). The Pt 4f$_{7/2}$ peak of PtRuP/carbon (1.2) (Figure 9) had a high binding energy shift compared to other samples (Figures 8 and 10). The PtRuP/carbon (1.2) sample had the highest amount of Pt$^0$ among all samples, consistently with the results in Table 5.

Since Pt should form an alloy with Ru to prevent poisoning, the electronic state of Ru was investigated. Most researchers have used Ru 3p to examine the electronic state of Ru, because Ru 3d peak is overlapped with C 1s. The Ru 3p peak intensity was weak, and the noise was significant. In this study, Ru 3d$_{5/2}$ peaks were studied. The peak at 280 eV was assigned to Ru$^0$; and the peak at 281 eV was assigned to RuO$_2$. All peak intensities were divided by ASF (atomic sensitivity factor), and C 1s peak intensity was used as a basis for comparison. The results in Table 5 show that the PtRuP/carbon (0.6) sample had the least amount of Pt$^0$ among all samples, PtRuP/carbon (2) had the second least, and PtRuP/carbon (1.2) sample had the highest amount of Pt$^0$. The PtRuP/carbon (1.2) sample also had the highest amount of Pt-RuO$_2$ on the surface among all samples. It contained 1.57% Pt and 1.33% RuO$_2$.

### 3.2.6. EXAFS

EXAFS was used to determine the state of Pt in alloy catalysts [25,26]. Table 6 summarizes the coordination numbers of Pt of the 60 wt.% PtRu-P/carbon black catalysts. N denotes the coordination number of Pt, and D is the metal particle size obtained from the TEM image. $\Delta E_0$ is the difference between experimental and theoretical values of bonding energy. $\Delta \sigma_j^2$ is the error of the thermal vibration of atoms. Figure 11 shows the Pt L$_{III}$ edge

$k^3$ of 60 wt.% PtRu-P/carbon black with various phosphorus content. It shows that all samples had a larger Pt-Pt distance than the theoretical value, which was obtained from the Pt/carbon black sample. This is because the intermixing of Pt and Ru atoms in the samples. It also confirms the formation of the Pt-Ru alloy. Figure 12 shows that the sample with a P/Pt atomic ratio of 1.2 had the smallest metal particle size and the lowest overall coordination number ($N_{Pt-Pt}$ + $N_{Pt-Ru}$) among all samples.

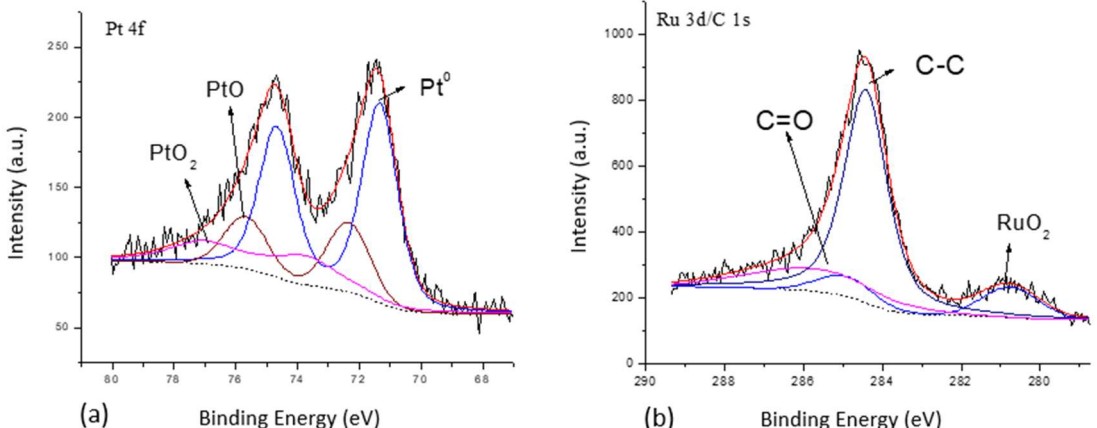

**Figure 9.** XPS spectra of PtRuP/carbon (1.2), (**a**) Pt 4f spectra; (**b**) Ru 3d/ C 1s spectra.

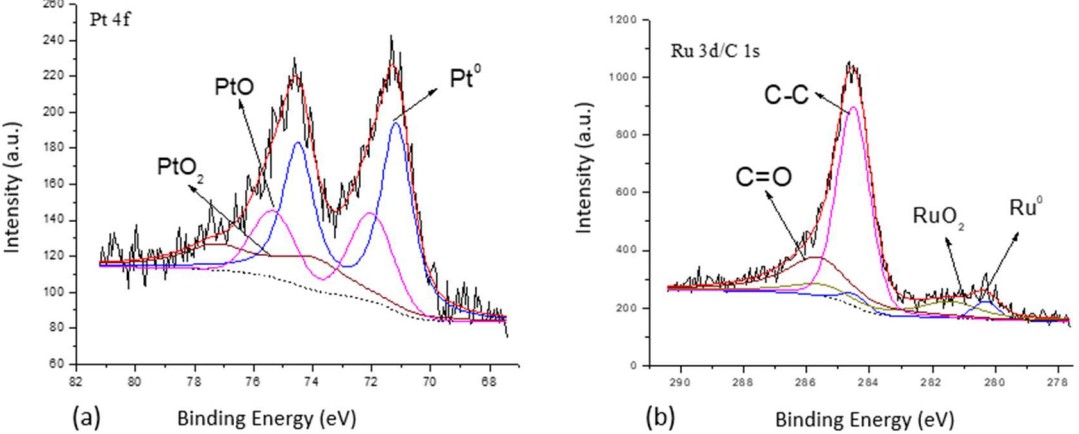

**Figure 10.** XPS spectrum of PtRuP/carbon (2), (**a**) Ru 3d/C 1s spectrum, (**b**) Pt 4f spectrum.

**Table 6.** Coordination numbers of various carbon black samples.

| Sample | Shell | N | Å | D (nm) | $\Delta E_0$ (eV) | $\Delta \sigma_j^2$ (A$^2$) | r-Factor |
|---|---|---|---|---|---|---|---|
| PtRuP/carbon (2) | Pt-Pt | 7.89 | 2.74 | 3.06 | 3.16 | 0.007 | 0.30% |
| | Pt-Ru | 1.66 | 2.69 | | 0.55 | 0.004 | |
| PtRuP/carbon (1.2) | Pt-Pt | 5.31 | 2.75 | 2.26 | 4.79 | 0.005 | 0.18% |
| | Pt-Ru | 2.03 | 2.71 | | 2.77 | 0.004 | |
| PtRuP/carbon (0.8) | Pt-Pt | 6.61 | 2.74 | 2.78 | 3.54 | 0.006 | 0.05% |
| | Pt-Ru | 1.54 | 2.7 | | 1.81 | 0.003 | |
| PtRuP/carbon (0.6) | Pt-Pt | 8.74 | 2.72 | 3.37 | 0.73 | 0.006 | 0.10% |
| | Pt-Ru | 0.62 | 2.70 | | 19 | 0.0006 | |

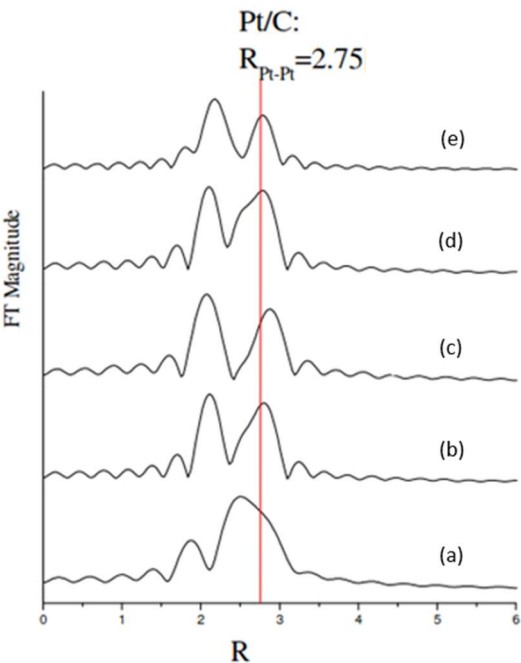

**Figure 11.** Pt $L_{III}$ edge $k^3$ of 60 wt.% PtRu-P/carbon black with various phosphorus content. (**a**) PtRuP/carbon (0.6), (**b**) PtRuP/carbon (0.8), (**c**) PtRuP/carbon (1.2), (**d**) PtRuP/carbon (2), (**e**) JM60 (JM60 stands for the sample Johnson Matthey 60 wt.% PtRu/C catalyst).

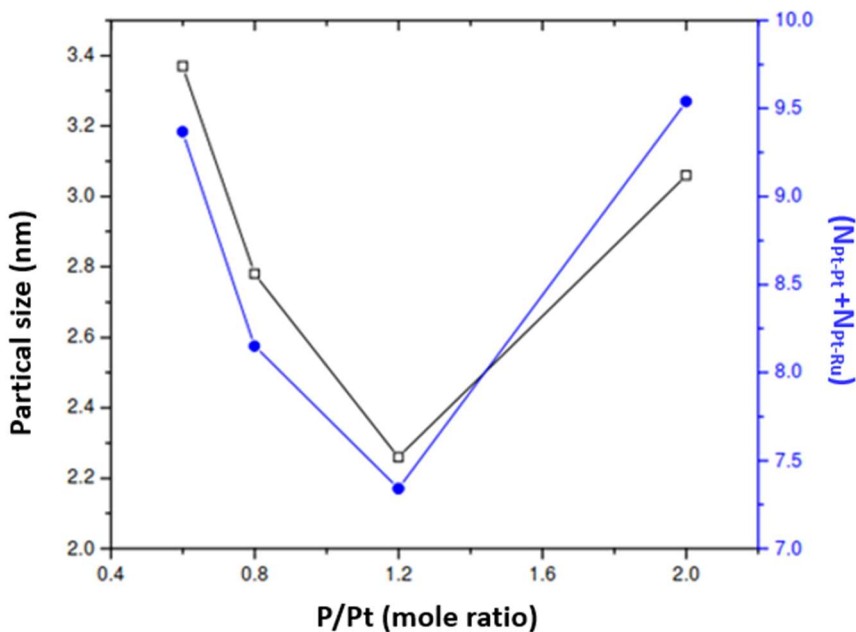

**Figure 12.** Metal particle size and ($N_{Pt-Pt}$ + $N_{Pt-Ru}$) versus P/Pt in the catalyst.

### 3.3. Catalytic Activity of the PtRu Catalysts

Table 7 shows that the sample reduced by $NaBH_4$ was 3.5 nm in size, that reduced by HCHO was 2.7 nm in size, and that reduced by $NaH_2PO_2$ was 2.5 nm in size. Their activities had a reverse trend. The activities were 84.95 A/g for the catalyst reduced by $NaBH_4$, 147.71 A/g for that reduced by HCHO, and 309.05 A/g for that reduced by $NaH_2PO_2$. In conclusion, the smaller the metal particle size is, the higher the activity is. The catalyst reduced by $NaH_2PO_2$ had the smallest particle size and the highest activity among all samples.

**Table 7.** Effects of reducing agent on the metal particle size and activity.

| Sample | Reduction Condition | | | | Particle Size (nm) | XRF (Atomic %) | | Loading (wt.%) | | Activity (A/g) |
| | Reducing Agent | Temp. | pH | Time (h) | TEM | Pt | Ru | Nominal | TGA Result | |
| --- | --- | --- | --- | --- | --- | --- | --- | --- | --- | --- |
| PtRu/carbon | NaBH$_4$ | RT | 12.1 | 2 | 3.5 | 63 | 37 | 59.9 | 57.72 | 84.95 |
| PtRu/carbon | HCHO | 85 °C | 12.2 | 3 | 2.7 | 61 | 39 | 60.1 | 46.17 | 147.71 |
| PtRu/carbon | NaH$_2$PO$_2$ | 90 °C | 11.3 | 10 | 2.5 | 57 | 43 | 59.8 | 55.02 | 238.2 |

\* Nominal metal ratio Pt:Ru = 50:50.

### 3.4. Effects of Phosphorus Content on the Activity

The CV curves of all the samples are shown in Figure 13. Table 8 shows the results of 60 wt.% PtRu-P/carbon black reduced by various amounts of NaH$_2$PO$_2$. The metal particle size of the sample decreased with an increase in the amount of NaH$_2$PO$_2$, consistent with that reported in the literature [19], i.e., the presence of P could suppress the growth of metal particles. The literature [19] did not mention why the metal particles became large when the P/Pt ratio was greater than 2. The XRD and TEM results show that the metal particle size of the sample decreased with an increasing amount of P. The TEM images also show that the metal particle size distribution was very uniform when the P/Pt atomic ratio was 1.2. However, the particles started agglomerate when the P/Pt ratio was 2. Table 7 shows that the activity decreased with an increase in metal particle size.

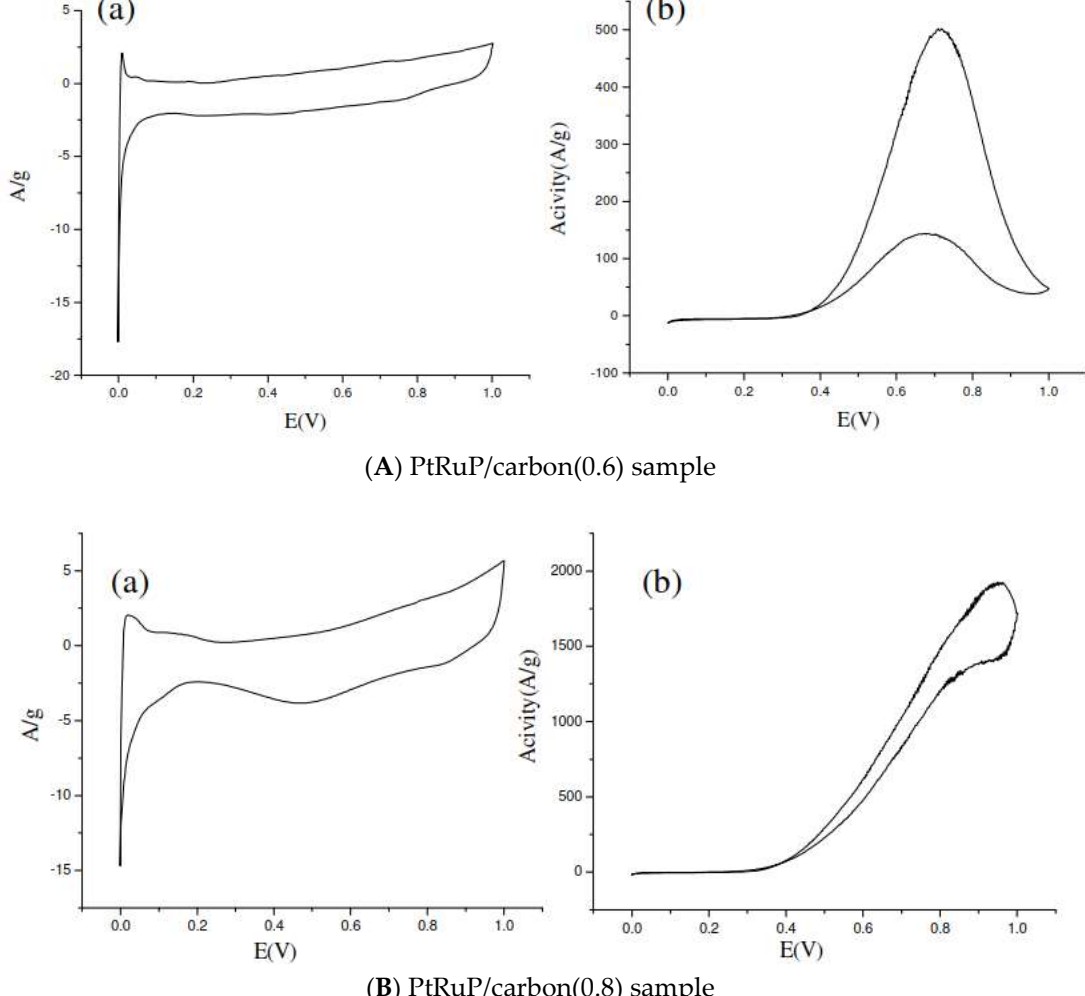

(**A**) PtRuP/carbon(0.6) sample

(**B**) PtRuP/carbon(0.8) sample

**Figure 13.** *Cont.*

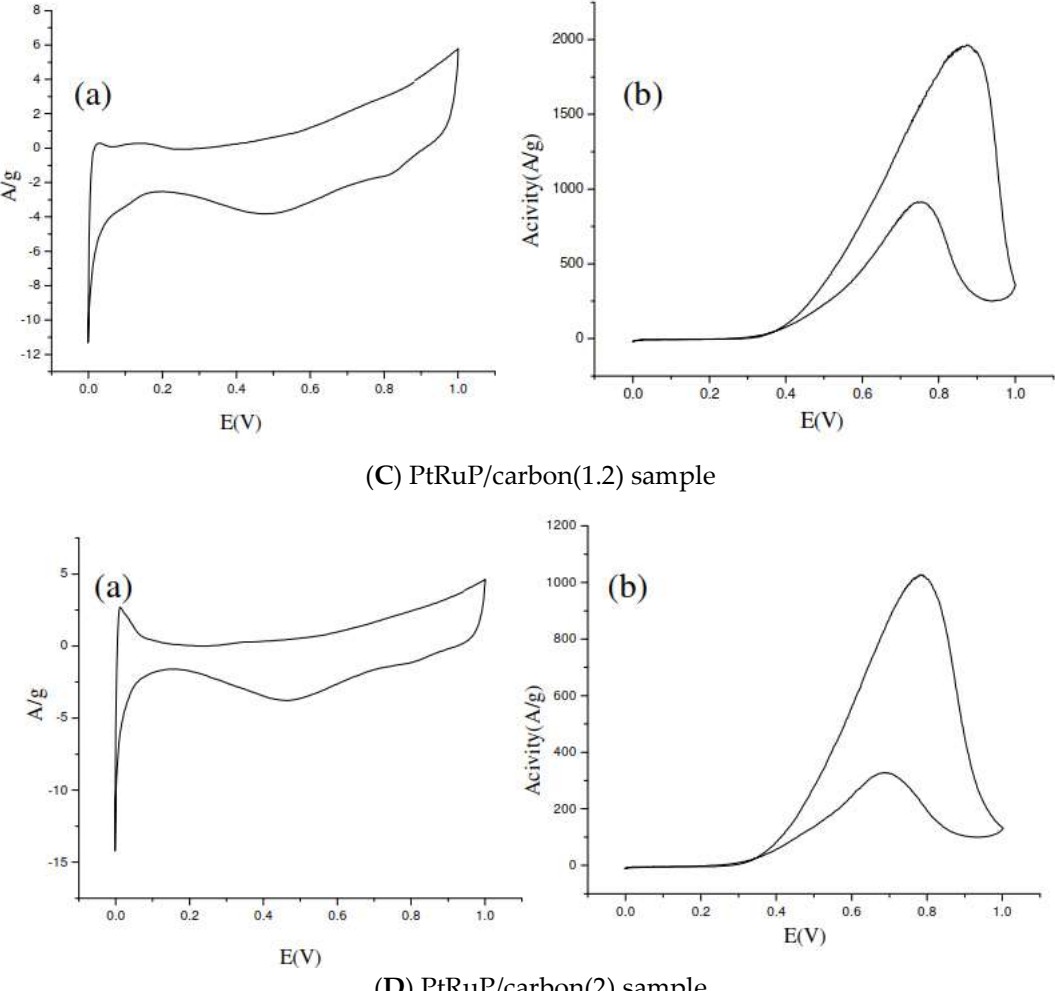

(**C**) PtRuP/carbon(1.2) sample

(**D**) PtRuP/carbon(2) sample

**Figure 13.** CV plots of the samples, (**a**) background of the 10th cycle, (**b**) activity of the 10th cycle.

**Table 8.** Effect of phosphorus amount on the activity.

| Sample | P/Pt | Metal Particle Size (nm) | | XRF | | Loading (%) | | Activity (A/g) |
|---|---|---|---|---|---|---|---|---|
| | | TEM | XRD | Pt (%) | Ru (%) | Nominal | TGA | |
| PtRuP/carbon (2) | 2 | 3.06 | 3.55 | 52 | 48 | 59.59 | 59.47 | 224.23 |
| PtRuP/carbon (1.2) | 1.2 | 2.26 | N.D. | 52 | 48 | 60.03 | 56.11 | 263.12 |
| PtRuP/carbon (0.8) | 0.8 | 2.78 | N.D. | 58 | 42 | 60.01 | 58.3 | 228.06 |
| PtRuP/carbon (0.6) | 0.6 | 3.37 | 3.48 | 54 | 46 | 60.08 | 57.77 | 89.9 |

The XPS results show that the PtRuP/carbon (0.6) sample had a small amount of $Pt^0$ (mole ratio 0.68%). Although its $RuO_2$ content was high (mole ratio 1.9%), methanol oxidation activity was mainly occurred on Pt. The PtRuP/carbon (1.2) sample's surface had the highest amount of $Pt^0$ (mole ratio 1.57%) among all samples. PtRuP/carbon (2) only had 1.16% (mole ratio) $Pt^0$. The results show that the higher the amount of $Pt^0$-$RuO_2$ is, the higher the activity is.

Figure 14 shows that when the P/Pt mole ratio was 1.2, the activity was the highest among all samples. When the P/Pt mole ratio increased up to 2, the activity dropped, since the particle size was large and the degree of the alloy was low. In conclusion, the 60 wt.% PtRu-P/carbon sample reduced by $NaH_2PO_2$ with a P/Pt ratio of 1.2 had very

high activity—its activity (263.12 A/g at 0.5 V) was higher than the commercial catalyst from Johnson Matthey with 60 wt.% PtRu/carbon (251.32 A/g).

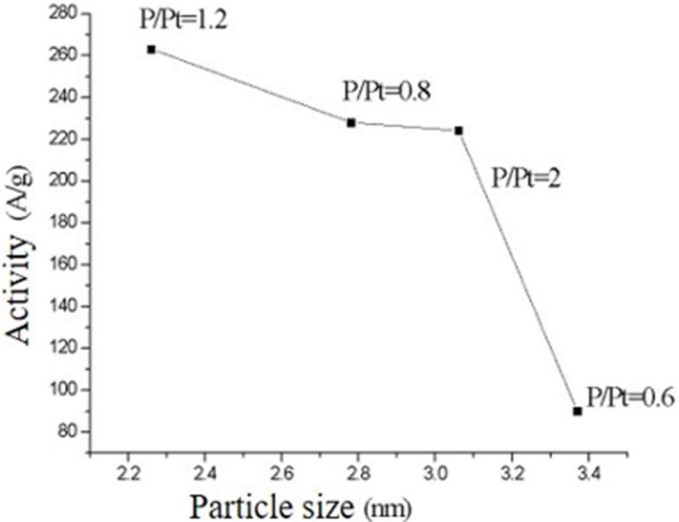

**Figure 14.** Catalyst activity vs. metal particle size.

One can calculate the fraction of the coordination number of Pt-Pt among all coordination numbers. The activity of catalyst vs. the fraction of the coordination number of Pt-Pt among all coordination numbers is shown in Figure 15. It shows that the activity increased as the fraction of the coordination number of Pt-Pt increased.

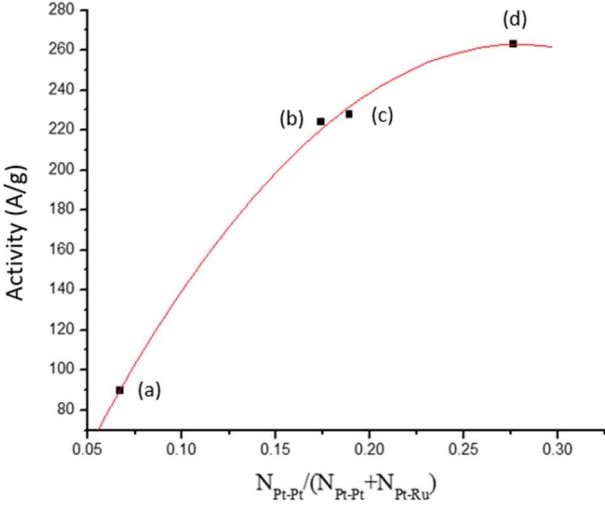

**Figure 15.** Catalyst activity **vs.** the fraction of coordination number of Pt-Pt among all coordination number. (**a**) PtRuP(0.6); (**b**) PtRuP(2); (**c**) PtRuP(0.8); and (**d**) PtRuP(1.2).

## 4. Conclusions

PtRu/carbon was reduced by $NaBH_4$, HCHO, and $NaH_2PO_2$, respectively. The sample reduced by $NaH_2PO_2$ had small metal particles and high methanol oxidation activity. When using $NaH_2PO_2$ as the reducing agent, the P/Pt ratio was crucial. Phosphorus would deposit on the surface of the carbon black, and suppressed the agglomeration of PtRu metal. The catalyst reduced by $NaH_2PO_2$ with a P/Pt raio of 1.2 had the highest activity among all catalysts. It had the higher $Pt^0$ and $Ru^0$ contents and smaller metal particle size than other catalysts. The carbon black-supported PtRu catalyst with $NaH_2PO_2$ as the reducing agent had the highest activity (263.12 A/g) among all catalysts in this study, which is higher than the commercial catalyst (Johnson Matthey H10100, 251.32 A/g).

**Author Contributions:** Conceptualization, Y.-W.C.; methodology, Y.-W.C.; validation, Y.-W.C. and H.-G.C.; formal analysis, H.-G.C.; investigation, Y.-W.C.; resources, Y.-W.C.; data curation, Y.-W.C.; writing—original draft preparation, Y.-W.C.; writing—review and editing, Y.-W.C. All authors have read and agreed to the published version of the manuscript.

**Funding:** This research was funded by Industrial Technology Research Institute, Hsinchu, Taiwan.

**Conflicts of Interest:** The authors declare no conflict of interest.

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
