# Peer review of "Effects of Reducing Agent on the Activity of PtRu/Carbon Black Anode Catalyst of Direct Methanol Fuel Cell"

_carbon, 2021_

Round 1
Reviewer 1 Report
In this manuscript, the authors developed a series of PtRu-based electrocatalysts for the methanol oxidation reaction. The authors carefully characterized the synthesized catalysts through TPD, XPS, XRD, TEM technologies. However, the data on the electrochemical part is not sufficient enough to support the high activity of catalysts. The investigation of the mechanism still needs to be described in detail. Overall, this manuscript is suitable to be published in this journal after the following revisions.
1. In the manuscript, the author should explain the mechanism of activity enhancement in detail. For example, the key effect of P during the catalytic process.
2. How about the stability of the as-prepared catalyst? Could the composition and structure be maintained stably after a long time operation?
3. Some data about the methanol oxidation activity of the catalyst are suggested to be provided, such as the CV curves.
4. The authors should cite more related references on MOR in the introduction and mechanism of the electrocatalytic process for the wider readership. Such as Appl. Catal. B: Environ. 282 (2021) 119595; Appl. Catal. B: Environmental 277 (2020) 119135; Small 2021, 17, 2007179.
Author Response
Reviewer I
In this manuscript, the authors developed a series of PtRu-based electrocatalysts for the methanol oxidation reaction. The authors carefully characterized the synthesized catalysts through TPD, XPS, XRD, TEM technologies. However, the data on the electrochemical part is not sufficient enough to support the high activity of catalysts. The investigation of the mechanism still needs to be described in detail. Overall, this manuscript is suitable to be published in this journal after the following revisions.
- In the manuscript, the author should explain the mechanism of activity enhancement in detail. For example, the key effect of P during the catalytic process.
Reply: NaH2PO2 could effectively reduce the particle size of PtRu metal. It can suppress the growth of metal particles. In addition, the P/Pt ratio is crucial. The catalyst reduced by NaH2PO2 with P/Pt raio of 1.2 had the highest activity among all catalysts. It had the higher Pt and Ru metal contents and smaller metal particle size than the other catalysts.
- How about the stability of the as-prepared catalyst? Could the composition and structure be maintained stably after a long time operation?
Reply: yes, the sample was tested for 200 h, it was still very stable.
- Some data about the methanol oxidation activity of the catalyst are suggested to be provided, such as the CV curves.
Reply: The CV curves are provided in the text.
- The authors should cite more related references on MOR in the introduction and mechanism of the electrocatalytic process for the wider readership. Such as Appl. Catal. B: Environ. 282 (2021) 119595; Appl. Catal. B: Environmental 277 (2020) 119135; Small 2021, 17, 2007179.
The following references have been added in the text.
- Yao, W.; Jiang, X.; Li, M.; Li, Y.; Liu, Y.; Zhan, X.; Fu, G.; Tang, Y. Engineering hollow porous platinum-silver double-shelled nanocages for efficient electro-oxidation of methanol, Appl. Catal. B: Environ. 2021, 282, 11959-11972.
- Li, Z.; Jiang, X.; Wang, X.; Hu, J.; Liu, Y.; Fu, G.; Tang, Y. Concave PtCo nanocrosses for methanol oxidation reaction, Appl. Catal. B: Environ. 2020, 277, 119135-119147.
- Li, M.; Li, Z.; Fu, G.; Tang, Y. Recent advances in amino-based molecules assisted control of noble-metal electrocatalysts, Small 2021, 17, 2007179-2007198.

Reviewer 2 Report
The authors provide the investigation of the PtRu/C catalyst for direct methanol fuel cells. They showed the effect of the reducing agent for preparing the PtRu/C catalysts. In addition, they studied the effect of the ratio of P/Pt in the reaction mixtures. However, the manuscript is not well organized and it seems just a description of the results only. Additionally, there are many points to be improved to publish in the journal as follows.
- The abstract should be more elaborate to well describe their results. In the last sentence, Pt0 and Ru0 can replace by Pt and Ru metals.
- In the experimental section, the first part of 2.4.2 is almost the same as 2.4.1. section. They wrote they conducted the electrochemical measurements at 24, 40 and 60 oC. However, there are no data in the manuscript.
- The sample naming of the sample is not well matched. Please, specify the P/Pt ratio in the PtRu/Cabon-NaH2PO2.
- The data in Table 4 and Figure 5 is not matched. Please, add the data from the PtRu/Carbon-NaH2PO2 if the P/Pt of the sample is 1.0 in Table 4 and Figure 5.
- The authors should reconsider the description in section 3.2.4. Figure 7 is not important to show in the manuscript. And the discussion for the XRD after TGA is not matched in the contexts.
- In Table 5, there is a sample of PtRu/carbon(1) at first. It should be defined or described in the experimental.
- In Table 7, the sample name is not defined previously.
- XPS data are described very redundantly. Please, make a concise description and summary for XPS data.
- In Figure 12, the data for P/Pt=1.0 should be included.
- The nitrogen adsorption and desorption isotherms should be provided in the manuscript or in the supplementary data.
Author Response
Reviewer II
The authors provide the investigation of the PtRu/C catalyst for direct methanol fuel cells. They showed the effect of the reducing agent for preparing the PtRu/C catalysts. In addition, they studied the effect of the ratio of P/Pt in the reaction mixtures. However, the manuscript is not well organized and it seems just a description of the results only. Additionally, there are many points to be improved to publish in the journal as follows.
- The abstract should be more elaborate to well describe their results. In the last sentence, Pt0 and Ru0 can replace by Pt and Ru metals.
We have revised the abstract. Pt0 and Ru0 has been replace by Pt and Ru metals.
- In the experimental section, the first part of 2.4.2 is almost the same as 2.4.1. section. They wrote they conducted the electrochemical measurements at 24, 40 and 60 oC. However, there are no data in the manuscript.
We have revised the experimental section, and the electrochemical measurements were carried out at 24 oC only.
- The sample naming of the sample is not well matched. Please, specify the P/Pt ratio in the PtRu/Cabon-NaH2PO2.
It is P/Pt= 1.
- The data in Table 4 and Figure 5 is not matched. Please, add the data from the PtRu/Carbon-NaH2PO2 if the P/Pt of the sample is 1.0 in Table 4 and Figure 5.
I have revised it. Thanks for your comments.
- The authors should reconsider the description in section 3.2.4. Figure 7 is not important to show in the manuscript. And the discussion for the XRD after TGA is not matched in the contexts.
Figure 7 has been deleted. The discussion for the XRD after TGA has been deleted.
- In Table 5, there is a sample of PtRu/carbon(1) at first. It should be defined or described in the experimental.
I have added it in the experimental section
- In Table 7, the sample name is not defined previously.
It has been added in the experimental section.
- XPS data are described very redundantly. Please, make a concise description and summary for XPS data.
The XPS data in Table has been revised.
- In Figure 12, the data for P/Pt=1.0 should be included.
I have revised the figure.
- The nitrogen adsorption and desorption isotherms should be provided in the manuscript or in the supplementary data.
Since only carbon black was measured for N2 sorption, and there are many figures in the text, I did not include it in the manuscript.

Reviewer 3 Report
In this work, the authors reported - PtRu/carbon black anode catalyst of the direct methanol fuel cell. The detailed studies of material characterizations have been extensively investigated. The experimental results of anode catalyst of direct methanol fuel cell were not extensively investigated. Overall, this work is not interesting, so I cannot suggest this present work to publish in the “Journal of Carbon Research”.
This manuscript could be improved by addressing the following issues.
- Authors should recheck the unique characters and using hyphen-minus, colon, superscripts and subscripts etc. (ex. Section 2.4.2 line – 4)
- There are many improper abbreviations that have been used. Authors should provide a proper way of presentation in order to understand the readers easily.
- Introduction part looks like an article abstract, so authors must elaborate on the contents related to the present work.
- Way of writing does not yet meet the journal format, where the characterization part is only focused by the authors; please try to focus all the necessary titles also with the same importance. And please make all the subsections of the characterization part into one title.
- In fig 3. Authors must write the plane details and should explain the importance of the image necessity.
- In fig 4 and 6. Authors must denote the scale bars in a larger size and produce high magnification HR-TEM images.
- Authors must include the XPS-deconvolution spectrum for Ru element.
- There is no evidence for electrochemical catalyst evaluation. Therefore, authors must produce cyclic voltagramms study.
Author Response
Reviewer III
In this work, the authors reported - PtRu/carbon black anode catalyst of the direct methanol fuel cell. The detailed studies of material characterizations have been extensively investigated. The experimental results of anode catalyst of direct methanol fuel cell were not extensively investigated. Overall, this work is not interesting, so I cannot suggest this present work to publish in the “Journal of Carbon Research”.
This manuscript could be improved by addressing the following issues.
- Authors should recheck the unique characters and using hyphen-minus, colon, superscripts and subscripts etc. (ex. Section 2.4.2 line – 4)
I have corrected all of them.
- There are many improper abbreviations that have been used. Authors should provide a proper way of presentation in order to understand the readers easily.
I have added the full names in the text.
- Introduction part looks like an article abstract, so authors must elaborate on the contents related to the present work.
It has been elaborated on the contents related to this study.
- Way of writing does not yet meet the journal format, where the characterization part is only focused by the authors; please try to focus all the necessary titles also with the same importance. And please make all the subsections of the characterization part into one title.
I have made all the subsections of the characterization part into one title
- In fig 3. Authors must write the plane details and should explain the importance of the image necessity.
In the XRD patterns in Figure 3, I have shown the plane in Figure 2.
- In fig 4 and 6. Authors must denote the scale bars in a larger size and produce high magnification HR-TEM images.
The scale bar in a large size has been added in Figures 4 and 6.
- Authors must include the XPS-deconvolution spectrum for Ru element.
I have shown the XPS deconvolution for Ru in Figure 10 (b)
- There is no evidence for electrochemical catalyst evaluation. Therefore, authors must produce cyclic voltagramms study.
The CV results have been added in the text.

Round 2
Reviewer 1 Report
The authors carefully addressed the comments and also performed experiments to improve the quality of the manuscript. The manuscript can be accepted for publication in its current form.
Author Response
Thanks to the reviewer 1. I highly appreciated your time and comments. It is very helpful.
Reviewer 2 Report
The authors responded to my comments on the previous manuscript. However, the quality of the manuscript is not improved well. They described they revised the experimental section for catalytic slurry for electrochemical measurements. Actually, however, they just deleted the title of the subsections. The corresponding author should be re-checked the contents of the manuscript. In addition, they did not revise Table 4 (table 3 in the revised manuscript) and Figure 5. These facts lead me to conclude the rejection of this revised manuscript.
Author Response
I revised the experimental section and cited my previous work, since the experimental section on for catalyst slurry and electrochemical measurement are the same as my previous work.
I also added the EXAFS results to show Pt and Ru formed alloy.
Reviewer 3 Report
In this work, the authors explained the Effects of Reducing Agent on the Activity of PtRu/Carbon Black Anode catalyst of Direct Methanol Fuel Cells. This article addressed methanol oxidation activity using PtRu/carbon reduced by NaBH4, HCHO and NaH2PO2 for direct methanol fuel cell applications. The detailed studies of material characterizations have been extensively investigated even though some significant issues should be addressed before being published in “Journal of Carbon Research.”
This article can be improved by addressing the following issues.
- Why do the authors add PdAu catalysts as a keyword?
- The authors should rewrite the Keywords alignment in the same format.
- In Section 2.2 preparation of anode catalyst, authors should provide abbreviations for chemical names.
- Authors should recheck for grammatical errors and spelling mistakes. (ex, section 2.4)
- In XRD analysis, authors should provide smooth XRD patterns.
- Authors have to insert the scale bars for TEM images.
- The authors should recheck for section 3.3 catalytic activity values using table 7.
- These works should be introduced to broader readers. They may consider several closely related references on PtRu/Carbon black anode catalyst in the methanol oxidation activity. It provides more information and enhances the quality and impact of the manuscript, such as
- Several closely related recent works need to cite in suitable place of the revised Introduction or Analysis parts.
Author Response
- Why do the authors add PdAu catalysts as a keyword?
It was a typo. We used PtRu as the key component of the catalyst. I have changed it to PtRu alloy.
- The authors should rewrite the Keywords alignment in the same format.
OK, we have done it.
- In Section 2.2 preparation of anode catalyst, authors should provide abbreviations for chemical names.
OK, we have done it.
- Authors should recheck for grammatical errors and spelling mistakes. (ex, section 2.4)
OK, we have done it.
- In XRD analysis, authors should provide smooth XRD patterns.
We provide the original XRD patterns, we think it is better than the smooth patterns. Besides, the XRD patterns we provided are quite smooth.
- Authors have to insert the scale bars for TEM images.
The scale bars were inserted in TEM images. We inserted the large size scale bars in this revision.
- The authors should recheck for section 3.3 catalytic activity values using table 7.
We have checked the values in Table 7, they are correct.
- These works should be introduced to broader readers. They may consider several closely related references on PtRu/Carbon black anode catalyst in the methanol oxidation activity. It provides more information and enhances the quality and impact of the manuscript, such as ceveral closely related recent works need to cite in suitable place of the revised Introduction or Analysis parts.
I added 3 recent papers:
- Yao, W.; Jiang, X.; Li, M.; Li, Y.; Liu, Y.; Zhan, X.; Fu, G.; Tang, Y. Engineering hollow porous platinum-silver double-shelled nanocages for efficient electro-oxidation of methanol, Catal. B: Environ. 2021, 282, 11959-11972.
- Li, Z.; Jiang, X.; Wang, X.; Hu, J.; Liu, Y.; Fu, G.; Tang, Y. Concave PtCo nanocrosses for methanol oxidation reaction, Catal. B: Environ. 2020, 277, 119135-119147.
- Li, M.; Li, Z.; Fu, G.; Tang, Y. Recent advances in amino-based molecules assisted control of noble-metal electrocatalysts, Small 2021, 17, 2007179-2007198.
There are many recent papers. It is impossible to add all of them in the manuscript. I do not know what references you want me to add.
Round 3
Reviewer 3 Report
Accept in present form